# Microbial Composition on Abandoned and Reclaimed Mining Sites in the Komi Republic (North Russia)

**DOI:** 10.3390/microorganisms11030720

**Published:** 2023-03-10

**Authors:** Aleksei O. Zverev, Grigory V. Gladkov, Anastasiia K. Kimeklis, Arina A. Kichko, Evgeny E. Andronov, Evgeny V. Abakumov

**Affiliations:** 1All-Russian Research Institute for Agricultural Microbiology (ARRIAM), 3 Podbelsky Chaussee, 196608 Saint Petersburg, Russia; 2Dokuchaev Soil Science Institute, Pyzhyovskiy Lane 7, 119017 Moscow, Russia; 3Applied Ecology Department, St. Petersburg University (SPbU), 7/9 Universitetskaya Emb., 199034 Saint Petersburg, Russia

**Keywords:** 16S rDNA amplicon sequencing, disturbed soil microbiota, northern soil microbiota, overgrown soils, quarry reclamation

## Abstract

Restoration of anthropogenically disturbed soils is an urgent problem in modern ecology and soil biology. Restoration processes in northern environments are especially important, due to the small amounts of fertile land and low levels of natural succession. We analyzed the soil microbiota, which is one of the indicators of the succession process is the soil. Samples were obtained from three disturbed soils (self-overgrown and reclaimed quarries), and two undisturbed soils (primary and secondary forests). Primary Forest soil had a well-developed soil profile, and a low pH and TOC (total organic carbon) amount. The microbial community of this soil had low richness, formed a clear remote cluster in the beta-diversity analysis, and showed an overrepresentation of *Geobacter* (Desulfobacteriota). Soil formation in clay and limestone abandoned quarries was at the initial stage, and was caused by both a low rate of mineral profile formation and severe climatic conditions in the region. Microbial communities of these soils did not have specific abundant taxa, and included a high amount of sparse taxa. Differences in taxa composition were correlated with abiotic factors (ammonium concentration), which, in turn, can be explained by the parent rock properties. Limestone quarry reclaimed by topsoil coverage resulted in an adaptation of the top soil microbiota to a novel parent rock. According to the CCA analysis, the microbial composition of samples was connected with pH, TOC and ammonium nitrogen concentration. Changes in pH and TOC were connected with ASVs from Chloroflexota, Gemmatimonadota and Patescibacteria. ASVs from Gemmatimonadota also were correlated with a high ammonium concentration.

## 1. Introduction

The Republic of Komi is one of the largest regions in Russia, and the largest one in the European part of the country. There are eight natural zones within the republic’s territory [1]. Ukhta city is located on the border of the middle and the northern taiga. The topography, lithology and bioclimatic conditions there are quite heterogeneous. In the last 70 years, soils of the republic have been investigated by numerous scientists in terms of their biology [2], soil organic chemistry [3] and classic cryopedology [4]. Although the region is large and has preserved many natural ecosystems, anthropogenic activity has led to an intensive transformation of the environment [5,6]. One of the significant factors involved in transforming the natural environment is open and shaft mining. Numerous quarries are located near Ukhta city. They are used for the mining of different resources, mainly construction materials (clays, sand and gravel, limestone, etc.).

The restoration of post-mining soils is a complex process that includes the management of all types of physical, chemical and biological disturbances of soils, such as the soil pH, fertility, microbial community, and various soil nutrient cycles [7]. Post-mining abandoned soils are normally characterized by undeveloped soil profiles, with an early stage of biogenic–abiogenic interactions occurring on the surface. The microbial composition of these soils is highly connected with abiotic factors. Many post-mining soils also contain significant amounts of toxic elements, or unnatural amounts of different chemical variables. Microbial communities of the soils are highly connected with soil properties. Alongside classical agrochemical analysis, they can be used for the description of restoration processes in soils. Unlike natural soils, the soil microbiota of post-mining soils shows a unique microbial composition. In terms of richness and evenness, the soil microbiota varies from rich to poor, mostly with hyper-variation in an abundance of different taxa [8].

The northern soils of Russia are relatively poorly studied. They are associated with both the severe climatic conditions of the region, as well as the weak development of the region. Despite this, the restoration of post-mining soils is an important problem. Due to climatic conditions, the restoration processes are relatively slow. This knowledge can be used for the greater understanding of soil restoration processes. The agrochemical data on primary soils formed in the quarries around Ukhta were published previously [9]. The microbiome of primary and natural soils around Ukhta city has not been studied enough, and only a few articles on soil biota have been published recently [10,11]. Single studies of the microbiome of soils and dumps in the cryolithozone of Russia include those investigating locations in Yamal [12,13] and central Yakutia [14]. In this regard, the study of the microbiome of post-anthropogenic soils and soils of the background (reference) ecosystems of the Komi Republic allows us to obtain more data regarding the microbial communities and processes occurring in complex northern soils. This work aimed to study the microbiome of soils of open pit dump complexes, as well as reference background soils—north taiga ecosystems—using modern methods of next-generation sequencing. The following objectives were set: (i) to evaluate the microbial diversity of soils, (ii) to identify microbial taxa, connected with differences in soil microbiota, and (iii) to measure abiotic factors and their influence on microbial biodiversity.

## 2. Materials and Methods

### 2.1. Sampling

Soils were described and sampled at different post-mining locations around Ukhta city, the Komi Republic, Russia (see Figure 1 and Table 1). Abandoned post-mining locations were presented by clay and limestone quarries (Dumped Limestone and Quarry Clay locations). Other locations were a limestone quarry, reclaimed by the top soil cover (Reclaimed Limestone), and a secondary forest nearby (Secondary Forest), which presumably was used as a topsoil source for the reclamation of the Reclaimed Limestone site. As a reference, the nearby location with the non-disturbed primary forest was selected (Primary Forest).

Samples were collected in five locations, with three replicas (sampled in 100 m proximities) in each location, for a total of fifteen samples. In every spot, the soil cut was set. The soil profile was described and the soil type was identified. Soil naming has been conducted according to WRB 2022 [15]. From the top soil (0–5 cm depth) of every spot, 500 g of soil mass was collected for the agrochemical analysis; 2 g of topsoil was sampled for the microbiological analysis (in 4 technical replicas). All soil samples were transported at +4 °C and stored at −20 °C. Information about sampling areas is provided in Table 1.

Despite the cold climate, there is no effect of permafrost on soil formation in the Ukhta region (in this region the permafrost zone is deeper than 1 m below the soil sampling level). The annual mean temperature is −1.05 °C, the mean July temperature is 15.9 °C, and the mean January temperature is −17.2 °C, with a total of 85 days having temperatures above 10 °C. The annual precipitation is 540 mm.

### 2.2. Agrochemical Analysis and DNA Extraction

The samples were air-dried, ground and passed through a 2 mm sieve. Soil solution was prepared in a ratio of 1:2.5 with water or 1M calcium chloride (CaCl_2_) [16]. The pH of the soil solution was measured by the pH meter pH-150MA (Belarus) [17]. Total organic carbon (TOC) content was determined by the Tyurin method, based on the oxidation of soil organic matter with a mixture of potassium dichromate and concentrated sulfuric acid [16]. The content of available forms of ammonium nitrogen (NH_4_^+^-N) and nitrate nitrogen (NO_3_^−^-N) was determined using a potassium chloride solution. The amount of free potassium and phosphorus was determined by the Kirsanov method [18].

The total soil DNA was isolated by using the MN NucleoSpin Soil Kit (Macherey-Nagel, Dueren, Germany), using a Precellys 24 homogenizer (Bertin, Montigny-le-Bretonneux, France) according to the manufacturer’s protocol. Quality control was carried out by PCR and agarose gel electrophoresis. The sequencing of the V4 variable region of the 16S rRNA gene was performed on the Illumina MiSEQ sequencer (Illumina, San Diego, CA, USA), using the primers 515f (GTGCCAGCMGCCGCGGTAA) and 806r (GGACTACVSGGGTATCTAAT) [19].

### 2.3. Bioinformatic Analysis

The general processing of sequences was carried out in R 4.0 [20], using dada2 (v. 1.14.1) [21] and phyloseq (v. 1.30.0) [22] packages, according to the authors’ recommendations. The 16S rDNA amplicon sequences were processed according to the dada2 pipeline. Sequences were trimmed by length (minimum 220 bp for forward and 180 bp for reverse reads) and quality (absence of N, maximum error rates maxEE were 2 for both forward and reverse reads). ASVs were determined according to the dada2 algorithm, and chimeric ASVs were removed by the “consensus” method. The taxonomic annotation was performed by the naive Bayesian classifier (provided in the dada2 package, default settings), with the SILVA 138 database [23] used as the training set; phyla names were corrected according to LPSN [24].

The alpha (observed ASV and Simpson indices)- and beta-diversity (Bray-Curtis distance) metrics were calculated using phyloseq and vegan [25] packages. The PCoA ordination of Bray–Curtis distances was drawn using the phyloseq package. The PERMANOVA analysis was carried out using the vegan package. The top 1000 most abundant ASVs were used for the CCA analysis in the vegan package [25]. The chemical variables used in the analysis were TOC, pH, P, K, N–ammonium and N–nitrate.

The differential abundance analysis was used for the determination of significant differences in the abundances of taxa between sites. This analysis was performed by the DeSEQ2 package [26]; the filters used were abs(log2FoldChange) ≥ 2 and baseMean ≥ 10.

## 3. Results

### 3.1. General Soil Characteristics

The mature soil was represented by a zone of Podzol, with a well-developed soil profile and signs of superficial humus (AY) and iron (BF horizon) illuviation. Podzol in the Ukhta city region had a full profile with clear eluvial–illuvial differentiation. Soils of the mines were represented by Leptosols (Clay Quarry samples), Leptosols with signs of gleyification (Secondary Forest samples) and Rendzic Leptosols (Limestone samples). Due to biogenic processes, the topsoil of the primary forest soil was characterized by an acidic pH and a relatively small amount of TOC.

According to primitive soil structure profiles, the soil development rate in the post-mining zones was relatively slow. It was caused by severe climatic conditions and a short period of biological activity. The mature soil was represented by a Podzol, with the feature of iron illuviation (in the BF horizon). Disturbed soils were represented by the initial soils that had formed on the mining heaps, of various genesis. Data on the chemical analysis of upper horizons are in Appendix A. All soils had a small percentage of humus. The level of soil acidity was higher in mature Podzols than in primary soils of the mining areas, due to the long impact of weathering and leaching in mature soils. The content of available forms of nitrogen was increased in humus and organogenic soil horizons when compared with other soil types. The results are presented in Table 2.

### 3.2. Soil Microbiota

After the bioinformatic processing was completed, 16,154 ASVs from 60 samples were obtained; the depth of sequencing ranged from 11,507 to 28,938 sequences per sample (11,507 after rarefaction for alpha-diversity analysis).

The alpha-diversity of samples is presented in Figure 2. According to the one-way ANOVA, for both observed ASVs (Df1 = 4, Df2 = 55, F = 13.59, *p* < 0.01) and Simpson (Df1 = 4, Df2 = 55, F = 9.93, *p* < 0.01) indices, the location was a significant predictor. The results of Tukey’s post-hoc tests are in Table 3.

The Primary Forest samples had the lowest value of richness, according to the observed ASVs index. It was significantly lower than the ones obtained for all other quarries’ samples, but not for the Secondary Forest. The richness of Secondary Forest samples was presumably higher than in Primary Forest samples and lower than in samples of quarries.

A significant difference in evenness (according to the Simpson index) had been discovered between the Primary Forest and Secondary Forest, as well as the Dumped Limestone and Quarry Clay, samples. Additionally, the evenness in Dumped Limestone samples was significantly higher than in Reclaimed Limestone samples. In general, the highest evenness was characterized for the disturbed quarry samples, whereas the more-developed soils had moderate values.

The results of the PCoA (beta-diversity, calculated using Bray–Curtis distances) are presented in Figure 3A. Due to the relatively high distances between the Primary Forest and other samples, the resolution of the PCoA plot for quarry samples was limited. The same plot was repeated without Primary Forest samples (Figure 2B). According to this data, the Primary Forest samples had a unique microbial composition, clustered in PCoA, in their own far cluster. Samples from the Reclaimed Limestone and Clay Quarry were also grouped in clear clusters, whereas Dumped Limestone and Secondary Forest samples had high inter-replica variation. According to the PERMANOVA, the sampling location was a significant factor (Df1 = 4, Df2 = 55, F = 16.38, *p* = 0.01) for the Bray–Curtis distances.

The taxonomic composition at the phyla level was typical for soils. Major phyla that were observed were Pseudomonadota, Bacteroidota, Acidobacteriota, and Actinomycetota, followed by Chloroflexota, Myxococcota and Verrucomicrobiota. There was no significant difference in the abundance of different phyla. According to the differential abundance analysis at the genus level, 128 ASVs were determined as variable taxa. Their relative abundances (in log transformation) are shown in Figure 4. ASVs are labeled at the genus level—in case of any absence of information, the lowest possible rank is provided.

The most abundant taxa from the soil microbiota of Primary Forest were *Paraphilimonas* (Bacteroidota), *Bradyrhizobium* (Pseudomonadota) and *Geobacter* (Desulfobacteriota). In Clay Quarry samples, the most abundant genera were *Niastella* (Bacteroidota) and *Variovorax* (Pseudomonadota), and in Dumped Limestone it was *Pseudarthrobacter* (Actinomycetota). Taxa from Secondary Forest and Reclaimed Limestone samples were not overrepresented.

The CCA revealed a connection between soil chemical variables and microbial ASV abundance (Df1 = 3, Df2 = 11, F = 2.55, *p* < 0.01). Significant predictors in the model were pH, total organic carbon (TOC) and ammonium concentration (N–ammonium). Using this model in the ordination of ASVs and samples, it was possible to determine how different ASVs or samples were affected by chemical factors. The ordination of ASVs is presented in Figure 5A. ASVs were annotated with different taxa levels, in search of a connection between the abundance of an ASV and the soil chemical factors. Three phyla-level taxa—Chloroflexota, Gemmatimonadota, and Patescibacteria—were estimated as being dependent on chemical factors. ASVs from Chloroflexota and Patescibacteria were connected with a high amount of TOC and a high pH, whereas ASVs from Gemmatimonadota were also connected with a low amount of ammonium nitrogen (Figure 4A). The sample ordination (Figure 5B) was determined by low TOC amount in the acidic soil of Primary Forest samples. Alongside this, the amount of ammonium nitrogen was also an important factor—Dumped Limestone soils had a low amount of nitrogen, whereas Quarry Clay samples had a high amount of it (Figure 5B).

## 4. Discussion

### 4.1. Primary Forest Soil Was Distinct from All Other Samples

Unlike all other soils in the dataset, the mature soil of Primary Forest samples was represented by Podzol, with a well-developed soil profile. The topsoil was characterized by an acidic pH and a relatively small amount of TOC. Microbial communities from the Primary Forest topsoil had their own specific structure (according to Bray distances) as well as a low richness. Such a specific microbial structure is typical for undisturbed primary forests, whereas a low richness is not [27]. Differential taxa analysis revealed several highly abundant taxa, associated with distinct microbial compositions. *Paraphilimonas* was previously reported as a novel strain from the rhizosphere of tomatoes [28], and was found in greenhouse soil in Korea [29]. A remarkably high amount of *Geobacter* suggests the specific role of this taxa in soil processes. *Geobacter* is known for its aerobic metal ion reduction in subsurface environments [30], so it is reasonable to suggest it being an intermediary in oxidation–reduction processes within soil. For instance, a significant amount of iron illuviation in the BF soil horizon can be associated with the metabolic activity of *Geobacter*. The other important taxa were *Puia* (Bacteroidota)—an aerobic bacteria located in broad-leaved forest soil in China, with an optimum pH of 4 [31]; *Bradyrhizobium* (Pseudomonadota); *Candidatus Udaeobacter* (Verrucomicrobiota), which is a widespread dominant group in acidic (pH ~ 5.1) soils [32]. A small amount of TOC can be explained by mostly copiotrophic communities, which are also typical in well-formed soils [33]

### 4.2. Secondary Forest Samples Were More Diverse Than the Primary Ones

The soil profile of the Secondary Forest sites was less developed than the one in the Primary Forest. Alongside a more neutral pH and increasing TOC amount, the microbial communities of the soil had a greater variation in both richness and the abundance of microorganisms. This is evidence of broader microbiological processes, associated primarily with local environment properties rather than with a specific role in soil metabolism. Significant taxa in this site were not overrepresented, and were generally associated with plants. For example, *Niastella* (Bacteroidota) was isolated from the rhizospheric soil of a persimmon tree [34]; *Skermanella* (Pseudomonadota) was previously found in meadow soil [35].

### 4.3. Abandoned Quarries Were Poor, but Different, Due to Their Soil Properties

Both Clay Quarry and Dumped Limestone samples were characterized by similar alpha-diversity—relatively high richness, followed by high evenness. Thus, the communities did not have specific abundant taxa, and were instead represented by a high number of sparse taxa. This is typical for disturbed, undeveloped soils, especially those in cold climates [14]. In contrast, the composition of taxa was different—according to beta-diversity, samples were separated into distinct clusters. According to CCA, this separation was highly correlated with abiotic factors (ammonium concentration in particular). This can be explained by parent rock properties. Clay soil resists water drainage and can accumulate different nutrition components, while limestone is easily drained.

Another difference was associated with the variation in replicas. Clay Quarries’ samples had the same microbial composition (which forms a clear cluster in beta-diversity), but varied in their richness. In contrast, Dumped Limestone samples had similar richness and evenness indices, but significant variations in terms of their taxa composition.

### 4.4. Limestone Recultivated Soils Had Their Unique Microbiome

Top soil is often used to supplement areas with poor substrate levels and to provide improved growth conditions [7]. The placement of a top soil cover (same as in the Secondary Forest site) over limestone parent rock led to a specific microbial community, distinct from both the Secondary Forest and Clay Quarry samples. These changes were the result of a well-developed topsoil microbial community adaptation to a novel parent rock. In spite of their high richness, the evenness in Limestone Recultivated samples was lower and closer to that from developed soils.

The relative abundant taxa analysis revealed taxa, previously known, from both the Secondary Forest (*Niastella*) and Dumped Limestone (*Pseudarthrobacter*) sites. Additionally, we identified *Bradyrhizobium* and *Rizobialis* (Pseudomonadota) taxa, having a range of ecological roles, including nitrogen fixation and bioremediation, as well as acting as plant pathogens [36].

### 4.5. pH, TOC and Ammonium Concentration Were Significant Abiotic Factors

According to the CCA analysis, pH, TOC and ammonium concentration were significant factors in the association of soil chemical variables with microbial ASV abundance. According to CCA, TOC and pH are environmental conditions that are associated with the unique composition of the Primary Forest samples, whereas the ammonium concentration factor had a more complicated effect. Alongside the unique composition of the Primary Forest samples, a low amount of ammonium was connected with Dumped Limestone Quarry samples, whereas a high amount was associated with the Clay Quarry. This is presumably linked to the substrate properties: clay soil resists water drainage and accumulates nutrition components (which boosts soil formation processes), while limestone is easily drained.

The CCA also revealed microbial taxa that are associated with these abiotic factors. Changes in pH and TOC were associated with ASVs from Chloroflexota, Gemmatimonadota and Patescibacteria phyla. Additionally, the presence of ASVs from Gemmatimonadota was correlated with high ammonium concentration. One of the Gemmatimonadota features is an adaptation to low soil moisture [37]. This supports the earlier hypothesis surrounding the connection between the water regime and ammonium concentration.

## 5. Conclusions

Soil formation in the abandoned quarries demonstrated a low mineral profile formation rate. This has been caused by both severe climatic conditions in the region and the poor nutritional content of materials exposed on the surfaces of the mines. The soil formation in these sites is in the initial stage.

Non-disturbed primary forest soil had a well-developed soil profile. It also was unique both in terms of is abiotic factors and microbial communities. Both a low pH and TOC were associated with a specific composition of the microbiota. The overrepresentation of *Geobacter* (Desulfobacteriota) was presumably connected with active oxidation–reduction processes in the soil. Secondary forest soils were less developed and had more variation in replicas.

Abandoned limestone and clay quarries were, primarily, undeveloped, primary soils. Microbial communities of these soils did not have specific abundant taxa, and instead included a high amount of sparse taxa. Differences in taxa composition were also connected with abiotic factors (ammonium concentration), which, in turn, can be explained by parent rock properties (in particular, the levels of the substrate). Recultivation of the limestone quarry by top soil coverage led to an adaptation of top soil microbiota to this novel parent rock, and to the creation of its own distinct microbiota.

The microbial composition of samples was hinged on three abiotic factors—pH, TOC (total organic carbon) and ammonium nitrogen concentration. Changes in pH and TOC were connected with ASVs from Chloroflexota, Gemmatimonadota and Patescibacteria. The presence of ASVs from Gemmatimonadota was also correlated with a high ammonium concentration.

## Figures and Tables

**Figure 1 microorganisms-11-00720-f001:**
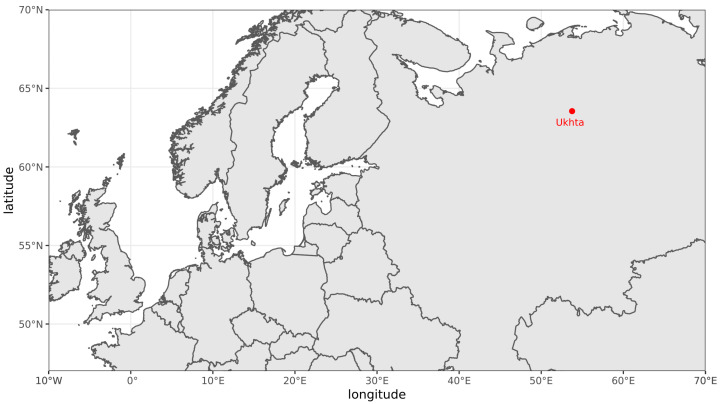
Large-scale depiction of the region utilized for sampling.

**Figure 2 microorganisms-11-00720-f002:**
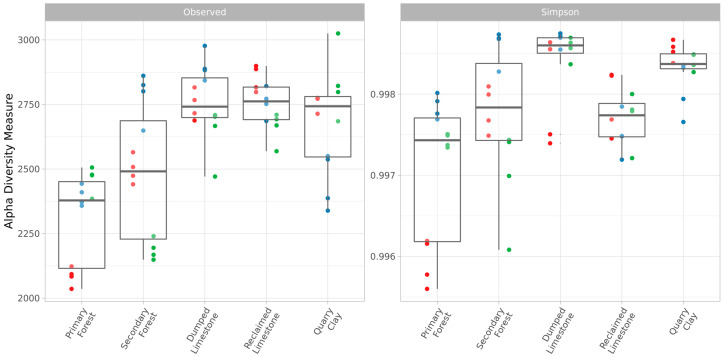
Alpha-diversity indices (observed ASVs and Simpson indices) of all samples. Different biological replicas are marked by color.

**Figure 3 microorganisms-11-00720-f003:**
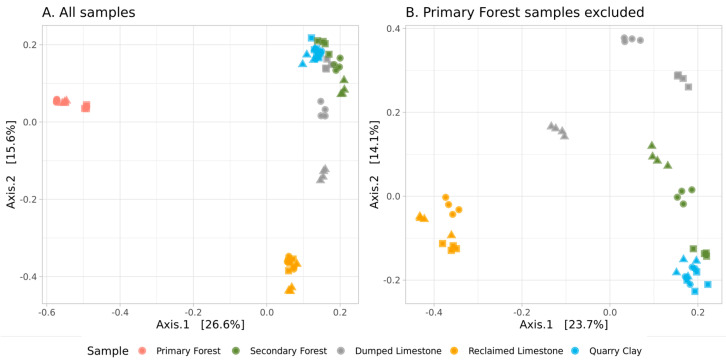
PCoA of Bray–Curtis distances between samples and replicas ((**A**)—all samples in the dataset; (**B**)—“Forest Global” samples are excluded). Different shape indicates different replicas.

**Figure 4 microorganisms-11-00720-f004:**
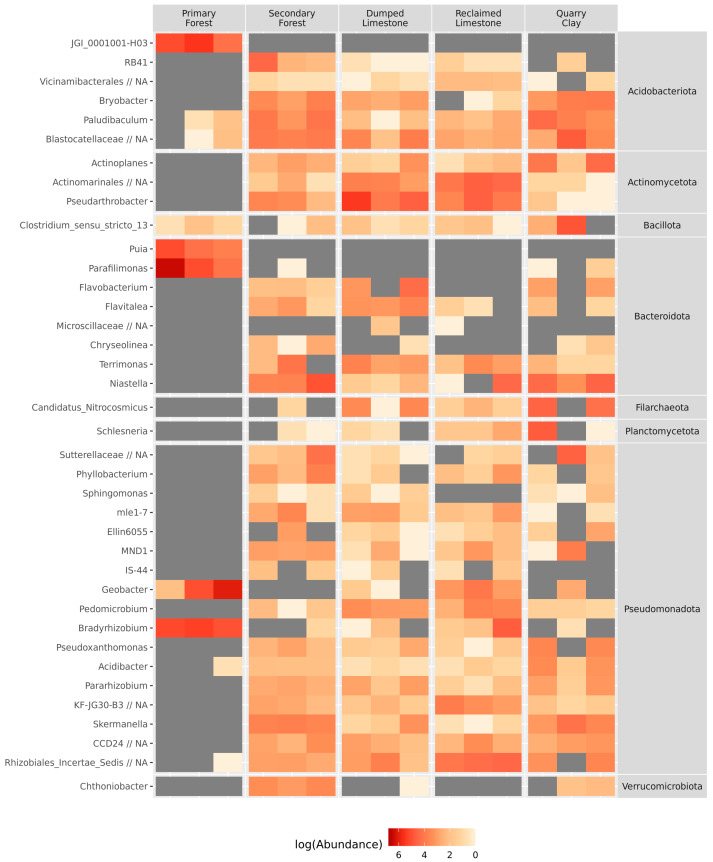
Relative abundances of significantly variable taxa (colored by log of abundance).

**Figure 5 microorganisms-11-00720-f005:**
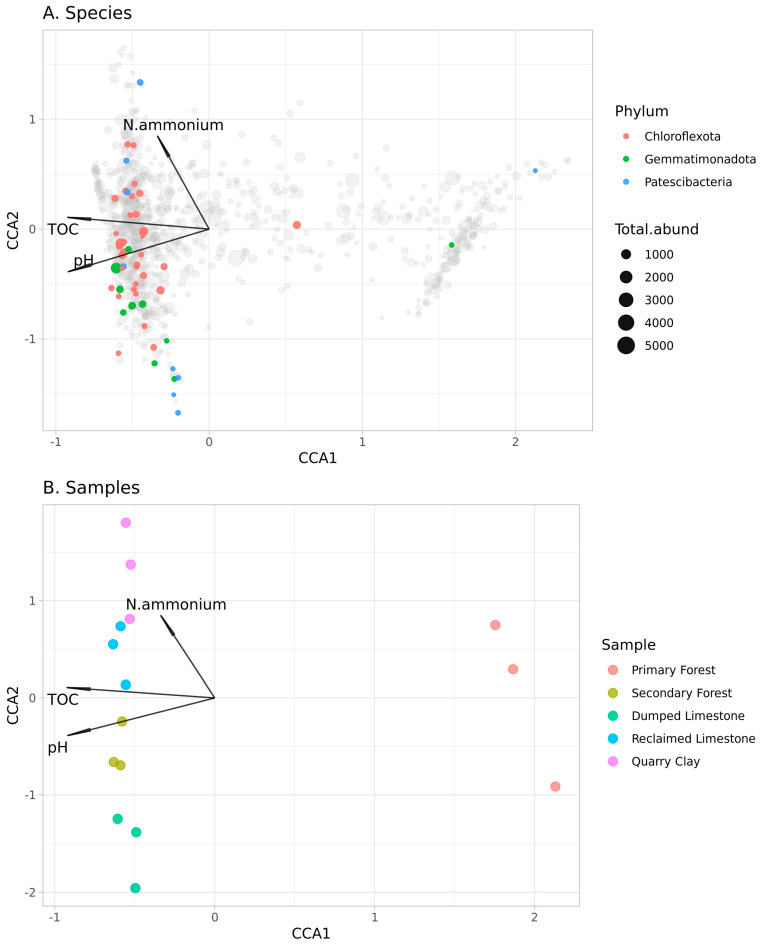
Visualization of the CCA analysis ((**A**)—ASVs and their total abundance, valuable phyla are highlighted by color (others are masked by grey colour), (**B**)—samples).

**Table 1 microorganisms-11-00720-t001:** Description of sampling locations.

Vegetation	Replica Details	Replica	Soil Profile	Location Details	Location
Clover, coltsfoot, cereals, St. John’s wort, moss	Bottom of the quarry uphill from water	I	W * C	Abandoned clay quarry near a road. Herbs overgrown on the bottom of the quarry	Quarry Clay N 63.54852, E 53.757625
Willow, sparse herbs, litter	Bottom of the quarry near the slope	II			
Grass cover, dotted spruce and pine	Bottom of the quarry near the water pond	III			
Pine, birch, rowan, moss, blueberry, wild rose, lingonberry, oxalis, legumes	Forest, forest litter	I	AY * E BF C	Primary pine forest with well-developed underwood	Primary Forest N 63.542693, E 53.776319
Pine, birch, club moss	Forest, forest litter and moss	II			
Birch, rowan	Forest, forest litter and grass	III			
Sedge, cereals, coltsfoot, legumes	Bottom of the quarry	I	W * AC Cc	Bottom of abandoned limestone quarry, broken stones with herbs	Dumped Limestone N 63.599297, E 53.776415
Birch, pine, willow, spruce, coltsfoot, Ivan-tea, crushed stone on the surface, moss	Bottom of the quarry, young forest	II			
Burdock, cereals, legumes, coltsfoot	Top of the local hill in the quarry	III			
Willow, pine, birch, cereals, legumes, Ivan-tea	Slope of dump	I	W * ACca Cg	Bottom of limestone quarry recultivated by topsoil cover, broken stones	Reclaimed Limestone N 63.589972, E 53.796547
Ivan-tea, moss, peas, willow, pine, birch	Slope with rocks	II			
Aspen, undergrowth of larch and pine, birch	Edge of quarry	III			
Undergrowth of pine, birch, willow. herbs, coltsfoot, clover, A moss, legumes, Ivan-tea	Edge of the forest	I	AU * AC Cca	Young secondary forest (pine, birch, willow)	Secondary Forest N 63.596806, E 53.782191
Birch, pine, moss, clover, legumes, dandelion, coltsfoot	Forest	II			
Pine, birch, willow, clover, legumes, Ivan-tea	Forest	III			

* marks the horizon of sampling.

**Table 2 microorganisms-11-00720-t002:** Measured soil chemical variables.

NO_3_^−^ mg/100 g	NH_4_^+^ mg/100 g	K, mg/100 g	P, mg/100 g	pH	TOC %	Replica	Location
15.2	98.6	369	42	6.8	1.45	I	Quarry Clay
1.59	44.3	337	7.3	7	1.56	II	
12.4	93	885	36	6.9	1.21	III	
2.5	7.62	59	46	6.4	0.66	I	Primary Forest
2.24	10.3	188	52	5.9	0.87	II	
0.77	19.2	296	67	5.5	0.55	III	
0.73	3.76	785	4.7	7.3	1.22	I	Dumped Limestone
1.38	9.36	118	6.8	7.5	1.34	II	
4.65	4.26	240	2.1	7.4	1.12	III	
10.6	13.4	205	3.4	7.3	1.45	I	Secondary Forest
1.08	17.5	188	1.3	7.2	1.65	II	
1.08	15.7	181	3.4	7.2	1.45	III	
8.52	19	77	0.9	7.2	1.65	I	Reclaimed Limestone
7.86	44.3	146	4.7	7.1	1.66	II	
5.24	37.9	216	3.4	7.1	1.7	III	

**Table 3 microorganisms-11-00720-t003:** Alpha-diversity: *p*-values for the Tukey’s post-hoc tests for the ANOVA results (upper triangle—observed ASVs, lower triangle—Simpson indices).

Quarry Clay	Reclaimed Limestone	Dumped Limestone	Secondary Forest	Primary Forest	
<0.01	<0.01	<0.01	0.137		Primary Forest
0.086	0.06	0.05		0.034	Secondary Forest
0.822	1		0.094	<00.1	Dumped Limestone
0.852		0.038	0.995	0.087	Reclaimed Limestone
	0.086	0.997	0.193	<0.01	Quarry Clay

## Data Availability

Not applicable.

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
