# Peer review of "Microbial Composition on Abandoned and Reclaimed Mining Sites in the Komi Republic (North Russia)"

_microorganisms, 2023, doi:10.3390/microorganisms11030720_

Round 1

Reviewer 1 Report

Major comment:

Due to prokaryotic taxa name validation in 2021, names of the taxa in the manuscript should be revised (see the valid publication: https://doi.org/10.1099/ijsem.0.005056) 

Several minor comments:

Abstract:

- Geobacter should be italic

- "connected", in my opinion sounds unclear. May be it should be replaced by "correlate"?

2.1 Section:

"sampled in 100m proximities" - space was missed

"The soil profile was described and the soil type was identified" - the used soil classification should be mentioned.

Author Response

We thank you for the review. All minor comments were improved. Soil database WRB 2022 was mentioned in the manuscript.

Unfortunately we were unable to access to the publication, recommended by you. Instead, names of the taxa in the manuscript have been revised by LPSN.

Reviewer 2 Report

GENERAL COMMENTS

The work entitled “Microbial composition on abandoned and reclaimed mining sites in the Komi Republic (North Russia)”

 RELEVANCE (considering the contribution to the advancement of knowledge): Good.

 ORIGINALITY (considering the problem to be studied and the existing knowledge gaps that justify the study): Good.

 TECHNICAL AND SCIENTIFIC MERIT: Good.

 FINAL OPINION: The work presents quality and merit and can be published. But it is necessary to attend to the following points:

What type of soil?

Soil classification according to the WRB. There is a new version of the 2022 WRB, please adapt.

Author Response

Thank you for the review. Soil classification was adapted to WRB 2022, type of soils were mentioned in the Results section